# The Possibility of Applying Acoustic Emission and Dynamometric Methods for Monitoring the Turning Process

**DOI:** 10.3390/ma13132926

**Published:** 2020-06-30

**Authors:** Krzysztof Dudzik, Wojciech Labuda

**Affiliations:** Faculty of Marine Engineering, Gdynia Maritime University, 81-225 Gdynia, Poland; w.labuda@wm.umg.edu.pl

**Keywords:** acoustic emission (AE), cutting forces, turning, cutting parameters, diagnostic, wear of cutting tool

## Abstract

Ensuring optimal turning conditions has a huge impact on the quality and properties of the machined surface. The condition of the cutting tool is one of the factors to achieve this goal. In order to control its wear during the turning process, monitoring was used. In this study, the acoustic emission method and measure of cutting forces during turning were used for monitoring that process. The research was carried out on a universal lathe center (CU500MRD type) using a Kistler dynamometer with assembled removable insert CCET09T302R-MF by DIJET Industrial CO., LTD. A dynamometer allows to measure forces Fx (radial force), Fy (feed force) and Fz (cutting force). The turning process was performed on a shaft with 60 mm diameter made of 304L stainless steel. The AE research was carried at Physical Acoustics Corporation with the kit that includes: recorder USB AE Node, preamplifier, AE-sensor VS 150M and computer with dedicated software used for recording and analyzing AE data. The aim of this paper is to compare selected diagnostic methods: acoustic emission and cutting forces measurement for monitoring wear of cutting tool edge. Analysis of the research results showed that both selected methods of monitoring the turning process allowed the determination of the beginning of the tool damage process.

## 1. Introduction

The finishing treatments used for the formation of shape and surface of a workpiece are e.g., turning, grinding, lapping, polishing, burnishing, etc. After treatment, the final dimensions as well as functional properties are imparted to a given element by application of proper treatment type [1]. Zhou et al. [2] observed that such effect will produce elements of high accuracy (3–5 accuracy class) and low values of roughness parameter Ra = 0.16 ÷ 0.01 µm.

Turning is still the most common method for surface layer forming. Conventional machining accuracy is usually considered as a function of the characteristics of all the components of object, tool, fixture and machine. Accuracy performance, the accuracy of static and dynamic determining and cutting parameters are associated with strength, temperature [3] and wear of the cutting edge. Lalwani et al. [4] investigated influence of cutting parameters on surface quality, and Balsamo et al. [5] observed the tool wear up to catastrophic failure, while Bhuiyan et al. [6] considered all those aspects. Those researchers used experimental tests results but Rao and Srinivas [7], on the basis of research results, developed a numerical method for predicting tool wear. 

Teti et al. [8] reviewed the current contribution of the CIRP (the French acronym of College International pour la Recherche en Productique-The International Academy for Production Engineering) organization to research on the development and implementation of monitoring of machining operation sensors, including tool condition monitoring, maintenance-free machining, process control and advanced topics in machining monitoring, innovative signal processing and related applications. Future challenges and trends in monitoring machining operations with sensors were presented and examples of their applications in industrial processes were given. The authors presented sensors used to monitor the machining process in various aspects. A review of the literature has shown that research centers are studying various aspects related to variable cutting conditions for machining various materials and production processes. Otieno and Abou-El-Hossein [9] investigated the machinability of aluminum, RSA 905, by changing cutting parameters in single-point diamond turning (SPDT). In order to optimize the machining process and determine tool wear, monitoring techniques were used by measuring forces and acoustic emission. The problem of tool wear was described, among others, by Yaman and Basaltin [10], who monitored the turning process of SAE 1030 steel on a CNC (Computerized Numerical Control) lathe for variable cutting conditions. Reddy T.S. and Reddy C.E. [11] attempted to predict tool wear and surface roughness by suggesting values and ranges of AE signals during monitoring in on-line systems. Aknouche et al. [12] used force measurement during wood milling to estimate cutting tool wear. Ensuring appropriate cutting conditions was described by Fukuzawa et al. [13], who used acoustic emission to verify the conditions of blade shape and thickness changes as well as the applied load to assess cutting conditions in the production of paper packaging. The research was aimed at developing an automatic system for detecting cutting conditions to apply precision machining for mass production. Schmidt et al. [14], who used a non-contact transducer to measure the detection and measurement of machine tool vibration for a single-point turning process in a production hall environment, was another example of testing cutting conditions that affect the achievement of the appropriate geometric surface structure. The research used workpieces made of M42 tool steel, tantalum alloys, 6061 aluminum alloy and Nitronic 33 stainless steel. These researchers proved in their research that obtaining a high surface quality depends on the optimized determination of cutting parameters and other interdependent factors.

Manufacturers of tools give ranges of recommended parameters, but in order to achieve the best possible surface area, it is necessary to define the exact parameters [15]. One of the criteria for the selection of turning performance is to reduce the tool vibration which results in surface quality, according to Filippov et al. [16]. This is particularly important for finishing treatment. Abbas et al. [17] observed that stabilization at the assumed level of the cutting force components allows to control the cutting tool deflection, which positively affects the quality of the machined surface. On the other hand, there are requirements for high machining efficiency in order to reduce production costs.

There are many methods for monitoring the machining process for selection of cutting parameters. One of them is the acoustic emission (AE) method [18].

According to the definition, acoustic emission is an evanescent elastic wave, which is the result of rapid release of the energy stored in the material by propagating a micro-damage (increase in micro-cracks, the movement of groups of dislocations) in the material or by a process (friction, leakage, etc.), confirmed in research of Babouri et al. [19]. The frequency range of a typical acoustic emission signal is usually determined in the range of 20 kHz to 2 MHz [20].

Acoustic Emission is considered as a passive non-destructive method. Its main advantages are:-the possibility of conducting continuous research,-high sensitivity of AE,-the possibility of carrying out research without having to shut down equipment out of service,-the possibility of locating the source of the AE signals generated by the cracks, leaks, etc.

The stimuli causing the release of energy and the formation of elastic waves can be: temperature change, environment, load operation and the processes which are accompanied by acoustic emission changes both at the micro and the macro scale, such as: friction, cracks [21], plastic deformation, leaks [22], corrosion, chemical reactions, structural and phase changes, delamination, cracking of the fibers and matrix in composites, etc.

The acoustic waves propagate in all directions from the source and can thus be recorded by one or more sensors fixed to an object or component. Due to the phenomenon of wave attenuation during their propagation, the distance from which they are detectable is limited. This distance depends on many factors, including material properties, object geometry and the level of interference from background noise, proved by Hase et al. [23]. 

Examples of AE signals are shown in Figure 1.

The AE signal can be characterized by parameters such as: number of exceedances of the threshold of discrimination hits, amplitude, duration, rise time, RMS and energy of the signal, etc. [25].

Many researchers have used AE for monitoring the turning process while Teti et al. [8] investigated additional milling and drilling processes. Bhuiyan et al. [26] used AE for monitoring turning and evaluating tool condition, while Al-Habaibeh et al. [27] applied additionally force measurement for that purpose. Because the tool condition has a great impact on treatment conditions and the same on quality of the product, that aspect was considered [28]. The tool condition is not only the wear stage but also built-up edge formation. All these phenomena and their influence on generating AE signals were considered by Ahmed et al. [29]. Albers et al. [30] on the basis on real time monitoring of AE proposed prediction of the product surface quality. Some researchers used another monitoring methods to achieve high quality and required surface roughness, i.e., Wu et al. [31] used AE, vibration and temperature measurement. Mikołajczyk et al. [32] propose an application of automatic detection of cutting edge wear on the basis of optical methods and artificial neural network in real-time monitoring systems. Li [33] presented the general state of knowledge on monitoring tool wear during turning by the AE method, summarizing achievements in this field up to 2002.

Despite the fact that works related to the topic of monitoring machining processes and tool wear has been carried out for many years, a large number of articles published testify to the validity of considering this issue. Some researchers focus on creating general principles of using various methods for monitoring machining processes and wear of tool. Others create mathematical models to help predict tool life or surface quality.

The aim of this study was to give exact values of chosen parameters indicating uncontrolled tool wear during turning in conditions described in the methodology. This article presents the possibility of using acoustic emission and dynamometer methods for monitoring turning process and wear of the tool.

## 2. Materials and Methods 

The research was carried out on a universal lathe center (CU500MRD, ZMM Sliven, Sliven, Bulgaria). The specimens were shafts with a diameter of 60 mm, made of 304 L stainless steel, with hardness of 215 HB. There were the same length of shaft pins–35 mm. The turning of the shaft was conducted by a tool with CCET09T302R-MF (DIJET Industrial CO., LTD, Osaka, Japan) removable insert. Basic information about the insert: C–rhombic insert shape with point angle 80°; C–insert clearance angle 7°; E–tolerance (nose height ±0.025 mm; thickness ± 0.025 mm; inscribed circle ±0.025 mm; T–insert type; 09-insert size = cutting edge length–9.525 mm; T3–insert thickness–3.97 mm; 02–nose radius–0.2 mm; R–right hand; MF–chipbreaker for finish turning of stainless steel. The typical cutting parameters for that insert are: cutting speed (Vc = 250 m/min), depth of cut (ap = 1.0 ÷ 2.0 mm) and feed (f = 0.05 ÷ 0.1 mm/rev). 

The tool was mounted into Kistler dynamometer (Kistler Instrumente AG, Winterthur, Switzerland). Dynamometer allowed measuring the cutting forces during turning process, both on conventional or CNC lathe machines. It consists of 9119AA2 piezoelectric dynamometer (Kistler Instrumente AG, Winterthur, Switzerland), 5070 charge amplifier and computer with DynoWare software (DynoWare type 2825D-02, Version 2.6.5.1.6, Kistler Group, Winterthur, Switzerland) for acquisition and analysing of the data. This dynamometer allows measuring of dynamic and quasistatic forces: Fx-radial force, Fy-feed force and Fz-cutting force. The range of measured forces is −4 ÷ 4 kN. The equipment measures the active force, regardless of the application point.

The view of tool and measuring equipment used in research is shown in Figure 2 and the view of shaft is presented in Figure 3.

On the basis of recommendations of the tool manufacturer, cutting parameters were selected. During this research, we used constant cutting speed Vc and changing feed f and depth of cut ap. The cutting parameters are presented in Table 1. 

The tests were carried out using one insert in the following order: ap = 1 mm and feed f = 0.05 mm/rev (1-st pin), f = 0.075 mm/rev (2-nd pin), f = 0.1 mm/rev (3-rd pin). Then the test was repeated for ap = 1.5 mm and ap = 2 mm. At the end of the test, a turning process was carried out using a new cutting insert-for comparison.

Monitoring of the turning process was conducted not only by cutting force measuring, but also by an additional acoustic emission method. Acoustic emission (AE) accompanying the turning process was recorded and analysed using a kit from Physical Acoustics Corporation. The kit includes single channel recorder USB AE Node, type 1283, preamplifier, AE-Sensor VS 150M, computer with AE Win for USB Version E5.30 software (Physical Acoustics Corporation, Mistras Group, Princeton Junction, NJ, USA).

The sensor (VS 150M, Vallen Systeme, Icking, Germany) was fixed to the surface of the dynamometer by means of a MAG4M magnetic holder-designed for that sensor. Silicone grease was used between the sensor and the surface as a coupling fluid. An overall view of the laboratory stand is shown in Figure 4.

The nose of cutting tool before and after every research stage was observed by digital measuring microscopy (Smartzoom 5, Zaiss, Oberkochen, Germany) Zaiss shown in Figure 5.

The chemical composition of the shaft material was analysed. It was carried out by an optical emission spectrometer (Solaris-ccd plus, GNR, Novara, Italy) with spark excitation, shown in Figure 6a. The view of the sample after four spark tests is presented in Figure 6b.

## 3. Results and Discussion

Cooperation with the industry and research confirms cases of differences in the chemical composition of materials supplied under different supplies. A relatively small change in the content of some alloying elements (e.g., Titanium or Sulfur) can significantly change the material properties affecting the durability of cutting tools during machining. Changing the turning conditions has a significant impact on the cutting forces and thus on the generated AE signals.

The results of the chemical composition testing of 304 L steel compared to the material’s certificate are presented in Table 2. Austenitic steels are the most important group of stainless steels. They have preferred combination of mechanical properties, machinability and corrosion resistance. However, they are classified as a group of materials difficult to machining process, according to Xavior et al. [34].

Chemical composition tests did not show significant differences compared to the certificate provided by the manufacturer of steel.

The results of the influence of cutting parameters on the measurement of Fz force are shown in Figure 7. The highest Fz mean value (582 N) was obtained for the following parameters: a cutting depth 2.0 mm and feed 0.075 mm/rev. For feeds of 0.05 and 0.075 mm/rev and an increase of depth of cut in all ranges, the Fz force increases too. For feed 0.1, while ap increases, the Fz force decreases.

Figure 8 presents the effect of changing such parameters as: ap and f on the feed force Fy. The highest Fy mean value (509 N) was observed for: a cutting depth of 2.0 mm and feed of 0.075 mm/rev. For other depths of cut, the mean values of feed force did not exceed 180 N. A significant increase of feed force testifies to an abrupt change of cutting conditions during the analyzed process.

Figure 9 shows the results of the influence of cutting parameter changes on the measurement of Fx force. The mean value of that force does not exceed −36 N for depth of cut in the range of 1.0–2.0 mm. The highest Fx mean value (−198 N) was obtained for: a cutting depth 2 mm and feed 0.075 mm/rev. A significant increase of radial force indicates a change in geometry of the cutting insert, and thus its damage or wear. This was confirmed during microscopic examination of the tool’s treating zone.

Table 3, Table 4 and Table 5 show the results of the statistical analysis of the Fx, Fy and Fz forces measurements. The highest standard deviation was obtained for ap = 2.0 mm and f = 0.075 mm/rev, and the maximum force reaches 1787 N.

The process of turning shaft pins with a cutting depth of 1.0 mm and an increased feed value resulted in a steady increase in the force Fz. Increasing the ap parameter to 1.5 mm contributed to a further increase in the analyzed force at 0.05 and 0.075 mm/rev feed. Increasing the feed rate to 1.0 mm/rev allowed to obtain the average value of Fz at the same level but can be observed as a wider spread of the results shown in Table 4. This may indicate less stable operation of the cutting tool. Another increase in cutting depth to 2.0 mm at a feed value of 0.05 mm/rev increases the average value of the cutting force, with a large spread of results, as evidenced by the value of the standard deviation (Table 5). Unfavourable cutting conditions which occurred in the previous stage of the study, contributed to the initial degradation of the cutting insert geometry, and during the turning process of the shaft pin with ap = 2.0 mm and f = 0.075 mm/rev of its damage. During the measurement of the force Fz for the cutting process, the difference between the minimum and maximum value of almost 1800 N was registered. Despite the damage to the cutting tool tip, the turning process carried out with an increased feed rate of f = 0.1 mm/rev ensured favourable conditions in the cutting zone and a decrease in the analysed force Fz to the average value. The tests carried out with selected cutting parameters in a similar way affected the values of radial and feed forces. For the cutting depths of 1.0 and 1.5 mm, the turning process was stable, but the increased value of the standard deviation and thus the deterioration of the cutting process conditions are also noticeable. Damage to the tip of the cutting insert caused several increases in the analysed forces Fx and Fy. In the case of Fz, a two-fold increase in value was observed.

The reason for such a clear increase of forces and their scattering is the damage of the nose of the cutting insert. The view of the cutting edge with different stages of wear can be seen in Figure 10. It shows the images of the rake face of the tool after individual passes.

In the first stage of cutting shaft pins (ap = 1.0 mm, f = 0.05, 0.075 and 0.1 mm/rev), only wiping on the surface layer of the cutting insert coating can be observed. The effects of the turning process can be seen only on the rake face, which corresponds to the depth of cut. Increasing the cutting depth to 1.5 mm at the same feed rates results in an additional loss of material at the corner radius and a slight deformation of the major cutting edge in the area of the rake face and flank face. The highest wear of the cutting tool occurred for the process carried out with ap = 2.0 mm. For a feed value of 0.05 mm/rev, a line is visible along the major cutting edge, which may indicate a crack in the insert coating on the chipbreaker. Visible damage in the form of abrasion on the flank face and notch, may contribute to faster abrasion of that surface, resulting in poor surface quality or dimensional inaccuracy of the detail. The insert’s wear process started when the feed was increased to 0.075 mm/rev. In the zone of the insert taking active part in the cutting process, the cutting edge was damaged along with breaking the entire corner radius with simultaneous wiping of the rake face and the auxiliary and main flank faces. Continued machining with a damaged insert caused the expansion of the degradation zone as well as thermal overheating and thermal cracks. In addition, micro chipping was observed, which also has a negative effect on surface finish and intensive wear of the flank face.

The analysis of forces during the turning process showed that, at the beginning of the cutting edge, the damage process can be observed for machining with cutting parameters ap = 2.0 mm and f = 0.05 mm/rev (Figure 11). Increasing the feed to 0.075 mm/rev caused damage of the cutting insert nose, in the form of a loss of part of the cutting edge, which affected the change of conditions during the cutting process. Cutting forces registered during the study were shown in Figure 12, while relatively large standard deviation obtained confirm the unstable cutting process. Increasing the feed value with a damaged cutting insert allowed to obtain a fivefold lower value of the Fz force and a significant reduction of the radial and feed force.

A further cutting process was performed with a cutting depth of 2.0 mm and a feed rate of 0.10 mm/rev, and despite the damage to the cutting insert, the analysed forces did not reach such significant values (Figure 13).

In order to properly carry out the turning process of stainless steel, it is important to ensure that in such cutting conditions, the chips produced during machining are short or in a controlled manner are discharged into the chip pan of lathe machine. Our work was carried out without the use of a cooling liquid, which also contributed to the formation of long chips. In Figure 11, Figure 12 and Figure 13, single or periodically repeating peaks can be observed. The formation of a long chip that is not properly removed remains in the direct cutting zone. Jamming of chips between the tool and the workpiece results in the generation of single or periodic increases in forces during the cutting process. This situation may cause the heated chip to cut again, and the situation may lead to thermal overheating and even chipping of the cutting edge of the tool.

In addition, a turning process was carried out using a new cutting insert with parameters: Vc = 250 m/min, ap = 2.0 mm and f = 0.1 mm/rev. Obtained results of measurements are presented in Table 6. The turning process carried out in this way resulted in the registration of the values of the analysed forces close to three times more than that for the insert with the damaged cutting edge.

An example of a graph recorded during the study, showing the change of cutting forces over time, is shown in Figure 14. The graph shows cutting forces during turning by a tool with a new cutting insert.

During the study, the acoustic emission (AE) signals generated in the lathing process were recorded. There are many parameters of AE. Chockalingam et al. [35] considered only amplitude in their investigations but the authors chose other patameters for further analysis i.e., amplitude, RMS, AE hits and frequency. The analysis of those parameters was made using AE Win for USB Version E5.30 software. 

Graphs obtained during the tests, for each of the feed values with constant cutting depth ap, are shown in Figure 15, Figure 16, Figure 17 and Figure 18.

In the final stage of turning with the parameters ap = 2.0 mm, f = 0.05 mm/rev, an increase in the amplitude and RMS of the recorded signal is noticeable. The RMS signal rises sharply to 50 µV; about twice as high compared to turning with a sharp cutting edge. Relatively low frequency of the signal and the increase of its amplitude (Figure 16) indicates a significant increase in its energy. This sudden increase in the energy of the generated AE signal may indicate the beginning of the destruction process of the cutting insert.

Figure 17 and Figure 18 show waveforms of signals recorded during turning with a cutting insert has a significant degree of wear. High RMS values (about 40–50 µV) of the signal indicate further damage of the insert. It is particularly visible in the initial turning phase with the parameters ap = 2.0 mm, f = 0.1 mm/rev where the signal RMS was recorded to be about five times higher than the undamaged insert; the maximum value was 111 µV. While the turning was performed in a good condition, this parameter was, on average, around 20 µV.

To calculate the mean values of the signal parameters, recorded during the test, the range characterized by the stability of the turning process was selected, omitting the initial phase of the tool approach to the workpiece and the final phase of the tool output from the material. The criterion of the reasonableness of the selection of measuring points was the duration of events, which were set at 75% of the maximum. The average values of selected parameters with their standard deviation are presented in Table 7.

For comparison, a signal generated during turning with a new insert was recorded with the parameters ap = 2 mm, f = 0.1 mm/rev.

An analysis of data recorded during the tests showed that it is possible to observe the beginning of a destruction of the nose of the cutting insert using the acoustic emission method. Figure 15 and Figure 16 show a sudden increase in the amplitude and RMS of the signal in the final stage of turning with the parameters ap = 2 mm, f = 0.05mm/rev. The tool failure process was continued while turning with the same insert with the parameters ap = 2 mm, f = 0.75 mm/rev. A signal with an average RMS value of about 70% higher than in the case of turning with previous parameters was recorded. Increasing the feed to f = 0.1 mm/rev at the same ap = 2 mm caused an increase in the RMS signal, with mean values being characterized by a large spread, as evidenced by a relatively large standard deviation.

A high RMS value and signal amplitude suggest further destruction of the tool. This is particularly evident in Figure 17 in the initial and final turning phases. Despite the significant damage of the nose of the cutting insert, the lathing process was stable. The relatively stable course of the process is evidenced by the decrease in the number of hits of recorded signal.

For comparison, a new insert was mounted with a sharp cutting edge and the turning was carried out with the parameters ap = 2 mm, f = 0.1 mm/rev. The character of the signal generated during turning with a new tool was similar to that recorded before the cutting edge of the insert was damaged. The number of hits is relatively low, whereas the RMS value of the signal was more than two times less compared to the signal generated during machining of the damaged tool, to the level of approximately 20 µV.

## 4. Conclusions

The machine tool operator should have, at his/her disposal, a device that will allow the determination of the wear of the cutting tool, quickly and accurately, as it has a significant impact on the quality of the machined surface, and thus on the performance of the surface layer. During the cutting edge operation period, the run-in period, normal wear and rapid degradation of the cutting edge can be distinguished. Many times, people who operate machine tools cannot precisely determine the moment of wear, e.g., of a cutting insert. The use of dynamometers for the analysis of forces during the cutting process makes it possible to observe changeable and unfavourable cutting conditions. The results presented in this article confirm the desirability of using a dynamometer for diagnosing the cutting process. Manufacturers of cutting tools propose a range of cutting parameters optimal for inserts, which should ensure stable cutting conditions with adequate durability. Damage to the cutting edge during the turning process can be observed by analysing the forces during machining the workpiece. Short-term changes in the limit values of registered forces may indicate chip jamming; therefore, they cannot be the basis for assessing the condition of the tool. A better solution can be achieved by observing the average force values. For stainless steel turning with CCET09T302R-MF insert, the indicator of the cutting tool corner damage may be an increase in the average value of the Fz force to the level of 600 N, while increasing the average value of the Fy force to 500 N and Fx to 200 N.

Widespread use of dynamometers may be limited due to the economic reasons of their purchase and the mounting problem for conventional lathes. In our case, a special tool holder was necessary for ensuring correctness of the machining process and registration of force measurements. Therefore, the use of the acoustic emission method to monitor the turning process can be an alternative that allows easy and quick measurement. This allows the observation of changing machining conditions or damage of the cutting tool.

Clear change in RMS of the AE signals (an increase of approximately 100%) could indicate damage to the edge of the turning tool. This increase was recorded with turning parameters: ap = 2.0 mm, f = 0.05 mm/rev. After rapid damage to the cutting edge of the tool in subsequent stages of the study, a stabilization of the process and a decrease of the signal RMS were observed, however, to a level higher than that for the unworn insert.

The maximum RMS value reached 111 µV while turning was performed with a new insert, this parameter was at the level of 20 µV. When the turning process was carried out with a worn out insert, despite the stabilization of the machining process, the average RMS value did not fall below 40 µV. In case of turning with the worn out insert, the recorded signal was not only characterized by a higher RMS value, but also by greater instability. During turning by a worn-out tool, the average value of standard deviation of RMS was many times higher, compared to the machining conditions with an unworn tool. This was probably due to the increase in the active length of the cutting edge, and thus the increase in cutting forces. The results allowed us to conclude that exceeding the RMS value above 40 µV may indicate excessive wear of the cutting tool during machining.

The subsequent stages of the tool cutting edge degradation process were observed by a digital measuring microscope (Zeiss Smartzoom 5). These tests confirmed the suspected degradation of the tool.

The research results of turning with the same cutting parameters, with a new insert compared to a worn-out one, showed that the technical condition of the tool has a significant impact on the generated AE signal, not the cutting parameters. This indicates the desirability of monitoring its condition. Studies have shown that both considered diagnostic methods: acoustic emission and cutting forces measurement can be useful during normal operation to detect excessive tool wear.

The works of other researchers, cited in the article, are often descriptive and general, while we focused on the practical aspect in conditions similar to our research (austenitic stainless steel with a hardness of about 215 HB (hardness of Brinell), a tool with similar geometry characteristic for finish treatment, lack of cooling, etc.). So far, we have completed basic research, which confirmed the sufficient accuracy and sensitivity of the selected methods. We tried to show that it is possible to monitor the process by direct (dynamometer) and indirect (AE) methods for obtaining relatively stable conditions of machining related to wear of the tool. We focused on showing exact parameters for the chosen material, tool and cutting parameters. The novelty is to provide specific RMS values of the AE signal as well as values of forces registered by dynamometer, indicating the beginning of rapid wear of the tool, in specific machining conditions, described in the research methodology.

## Figures and Tables

**Figure 1 materials-13-02926-f001:**
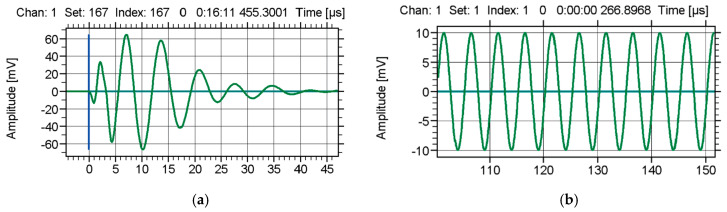
Examples of typical acoustic emission signals: (**a**) burst signal, (**b**) continuous signal [24].

**Figure 2 materials-13-02926-f002:**
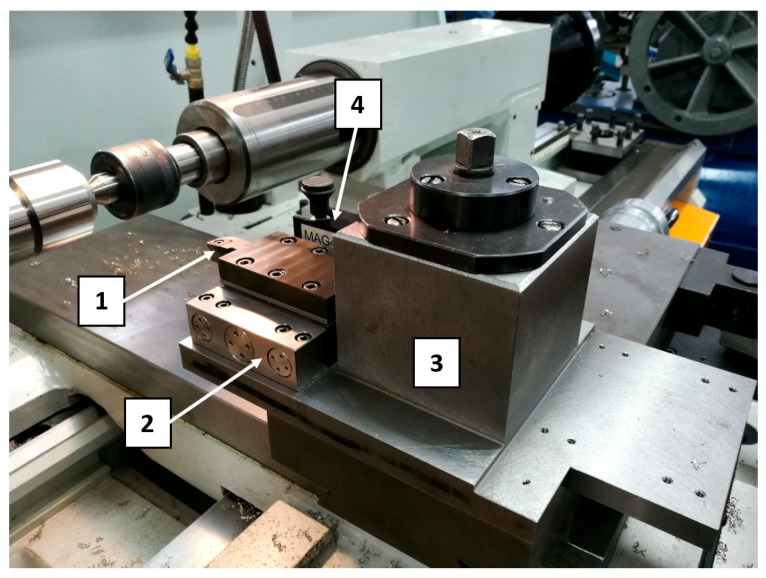
The equipment used in research: 1-tool, 2–dynamometer, 3–dynamometer grip, 4–acoustic emission sensor with holder.

**Figure 3 materials-13-02926-f003:**
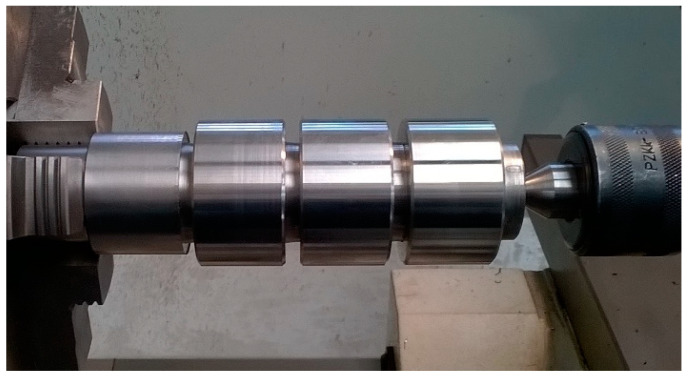
The shaft used in research.

**Figure 4 materials-13-02926-f004:**
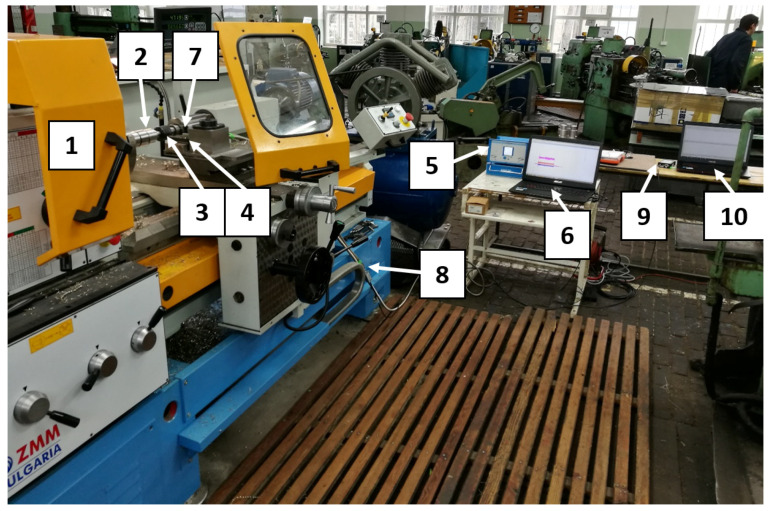
Laboratory stand: 1–lathing machine, 2–shaft, 3–tool, 4–dynamometer, 5–dynamometer recorder, 6–dynamometer computer, 7–AE sensor, 8–preamplifier, 9–AE recorder, 10–AE computer.

**Figure 5 materials-13-02926-f005:**
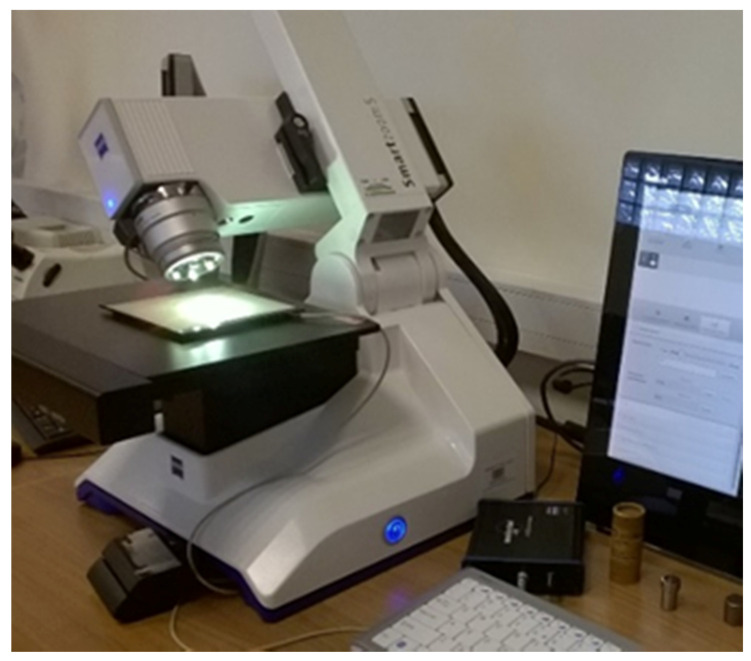
Measuring microscope used for observing wear degree of cutting insert.

**Figure 6 materials-13-02926-f006:**
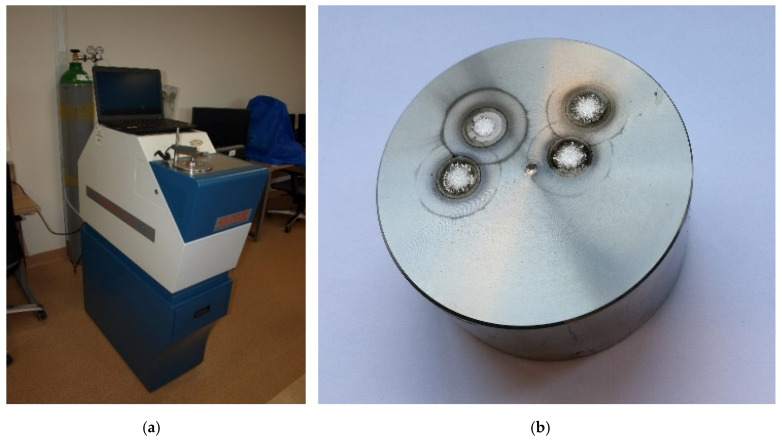
An optical spectrometer used for chemical composition test (**a**) and the view of the sample after four spark test (**b**).

**Figure 7 materials-13-02926-f007:**
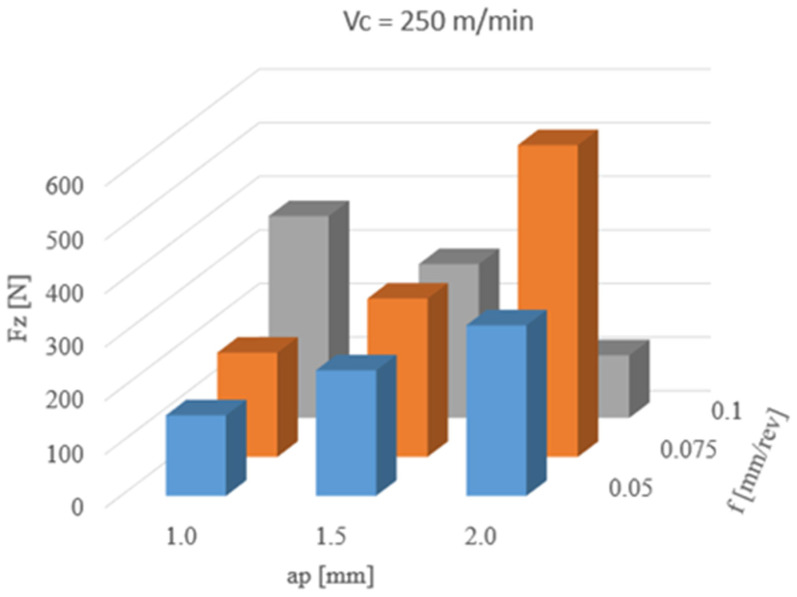
The results of changing cutting parameters on Fz.

**Figure 8 materials-13-02926-f008:**
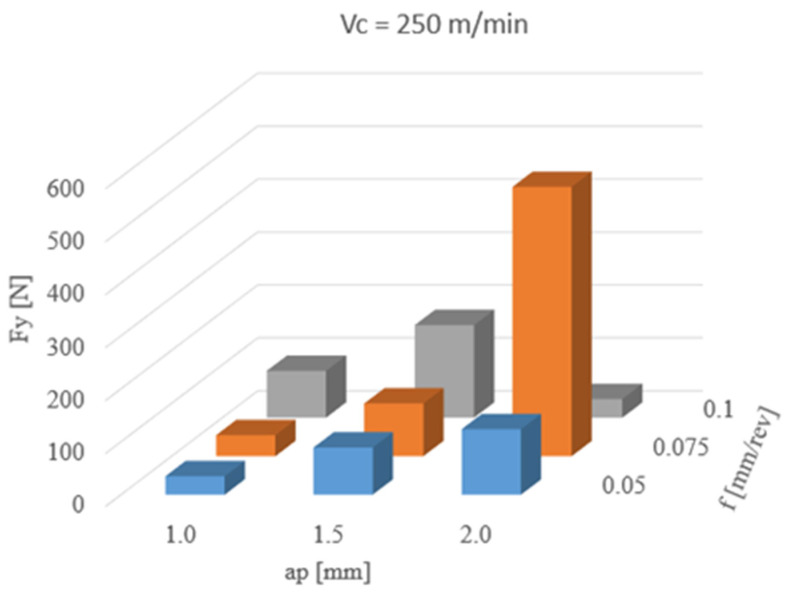
The results of changing cutting parameters on Fy.

**Figure 9 materials-13-02926-f009:**
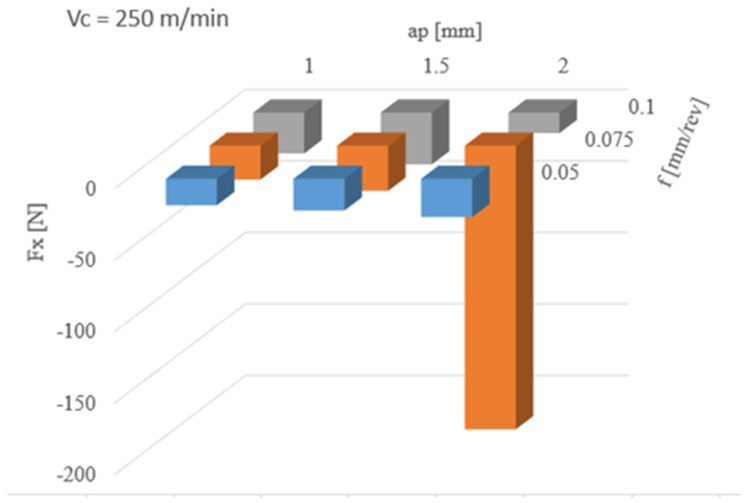
The results of changing cutting parameters on Fx.

**Figure 10 materials-13-02926-f010:**
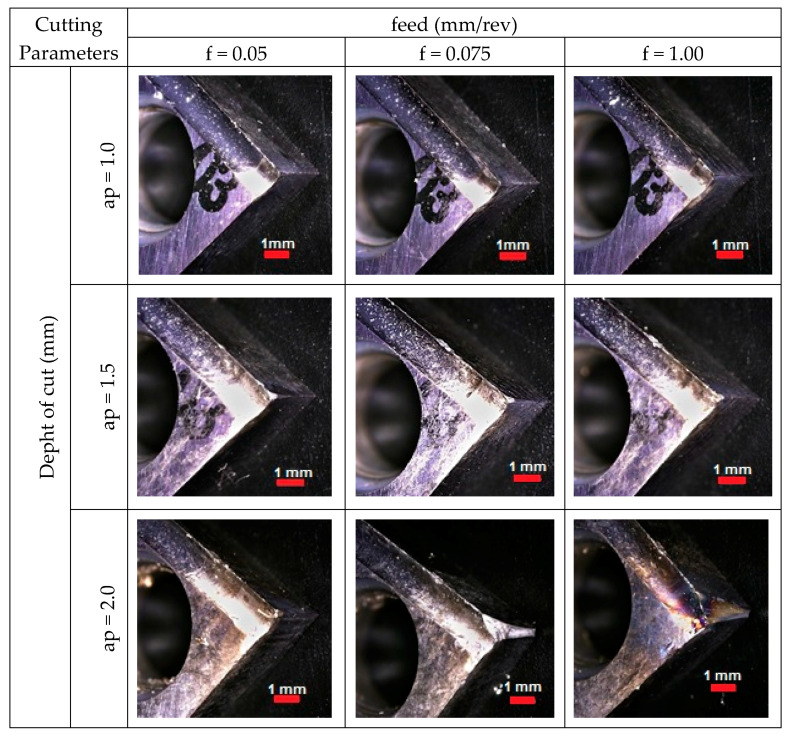
The wear of cutting edge.

**Figure 11 materials-13-02926-f011:**
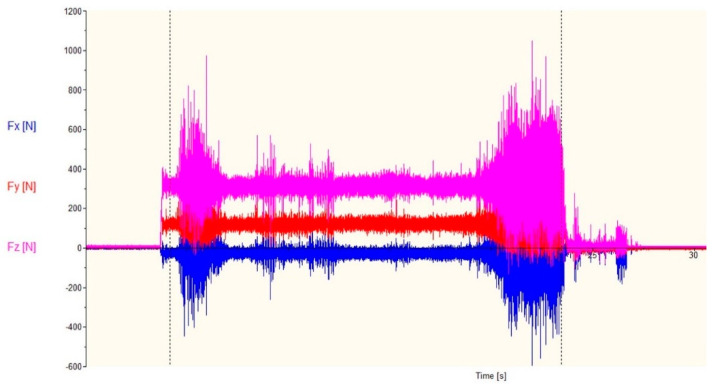
Cutting forces during turning with parameters: ap = 2.0 mm, f = 0.05 mm/rev.

**Figure 12 materials-13-02926-f012:**
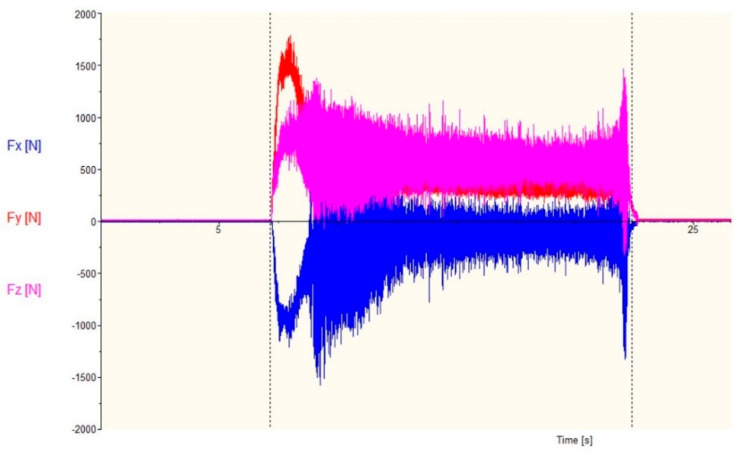
Cutting forces during turning with parameters: ap = 2.0 mm, f = 0.075 mm/rev.

**Figure 13 materials-13-02926-f013:**
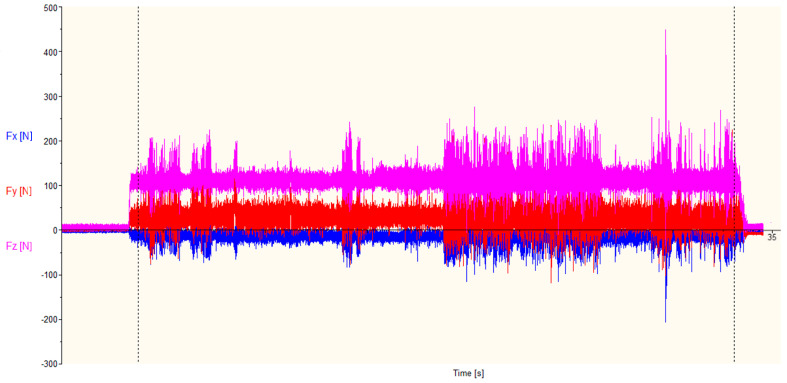
Cutting forces during turning with parameters: ap = 2.0 mm, f = 0.10 mm/rev.

**Figure 14 materials-13-02926-f014:**
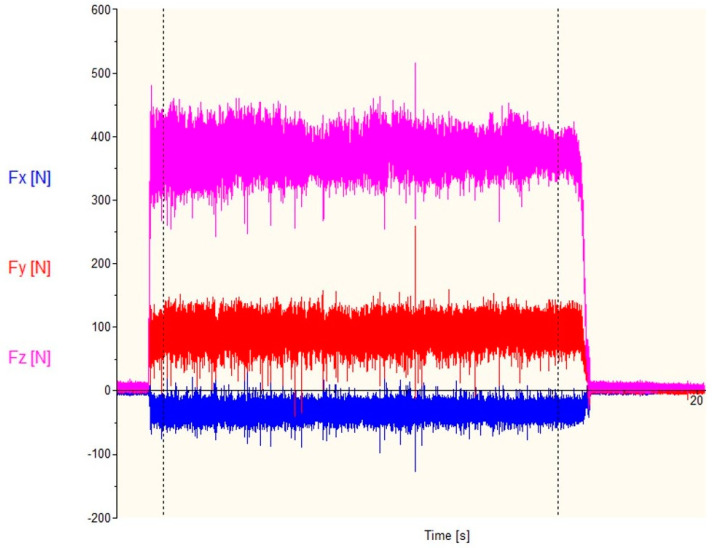
Cutting forces during turning by tool with new cutting insert with parameters: ap = 2 mm, f = 0.1 mm/rev.

**Figure 15 materials-13-02926-f015:**
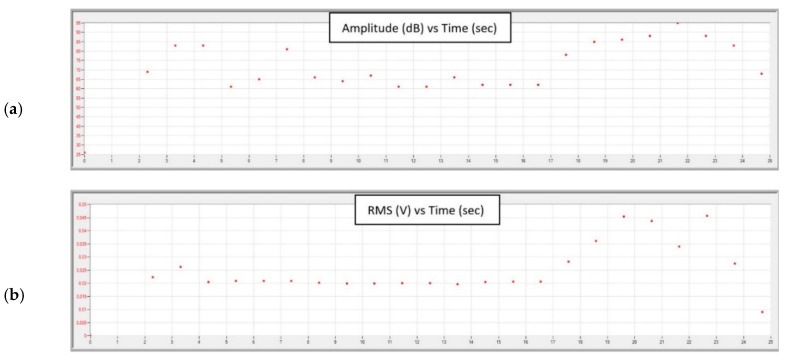
Charts of (**a**) amplitude and (**b**) RMS changing as a function of time during turning with parameters: ap = 2.0 mm, f = 0.05 mm/rev.

**Figure 16 materials-13-02926-f016:**
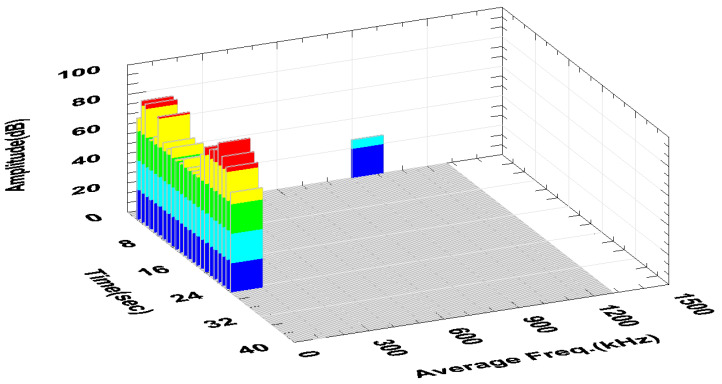
Chart of amplitude and average frequency changing as a function of time, during turning with parameters: ap = 2.0 mm, f = 0.05 mm/rev. Different colours help to observe changes in amplitude-every 20 dB.

**Figure 17 materials-13-02926-f017:**
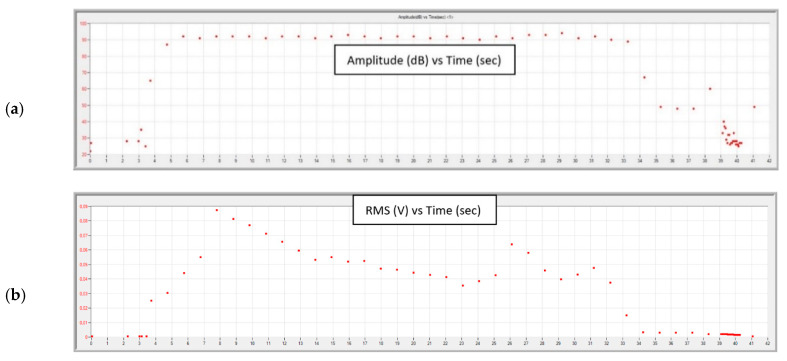
Charts of (**a**) amplitude and (**b**) RMS changing as a function of time during turning with parameters: ap = 2.0 mm, f = 0.075 mm/rev.

**Figure 18 materials-13-02926-f018:**
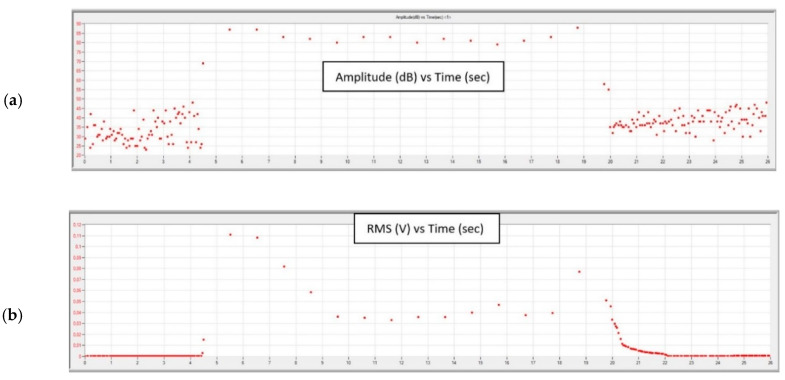
Charts of (**a**) amplitude and (**b**) RMS changing as a function of time during turning with parameters: ap = 2.0 mm, f = 0.1 mm/rev.

**Table 1 materials-13-02926-t001:** The cutting parameters used in research.

Cutting Parameters	Values
Vc (m/min)	250
f (mm/rev)	0.05; 0.075; 1.0
ap (mm)	1.0; 1.5; 2.0

**Table 2 materials-13-02926-t002:** Comparison of the measurements results of the chemical composition with the material certificate.

Data Source	Chemical Composition (%)								
	C	Si	Mn	P	S	Cr	Mo	Ni	Nb
measurement	0.037	0.457	1.638	0.028	0.030	18.261	0.473	7.760	0.008
certificate	0.018	0.46	1.74	0.031	0.030	18.12	-	8.02	-
	**Al**	**Cu**	**Co**	**B**	**Ti**	**V**	**W**	**Fe**	
measurement	0.003	0.483	0.125	0.002	0.026	0.057	0.021	70.594	
certificate	-	-	-	-	-	-	-	-	

**Table 3 materials-13-02926-t003:** Statistical analysis of cutting forces for ap = 1 mm.

ap = 1 mm									
f (mm/rev)	0.05			0.075			0.1		
Cutting Forces	Fx	Fy	Fz	Fx	Fy	Fz	Fx	Fy	Fz
	(N)	(N)	(N)	(N)	(N)	(N)	(N)	(N)	(N)
Mean	−19	35	150	−24	40	195	−29	89	376
Minimum	−114	−72	41	−162	−104	52	−316	−179	133
Maximum	93	159	238	125	238	346	192	660	826
Stand. dev.	6.1	8.0	9.8	13.1	15.4	17.1	10.5	17.3	24.2
Stand. error	0.02	0.02	0.03	0.03	0.04	0.04	0.03	0.04	0.06

**Table 4 materials-13-02926-t004:** Statistical analysis of cutting forces for ap = 1.5 mm.

ap = 1.5 mm									
f (mm/rev)	0.05			0.075			0.1		
Cutting Forces	Fx	Fy	Fz	Fx	Fy	Fz	Fx	Fy	Fz
	(N)	(N)	(N)	(N)	(N)	(N)	(N)	(N)	(N)
Mean	−22	89	234	−32	100	296	−36	175	286
Minimum	−71	44	122	−234	26	56	−617	−72	−239
Maximum	34	149	303	158	361	582	466	574	1044
Stand. dev.	6.7	8.5	13.4	10.9	11.6	16.1	83.3	56.5	96.8
Stand. error	0.01	0.02	0.03	0.03	0.03	0.04	0.15	0.10	0.18

**Table 5 materials-13-02926-t005:** Statistical analysis of cutting forces for ap = 2.0 mm.

ap = 2 mm									
f (mm/rev)	0.05			0.075			0.1		
Cutting Forces	Fx	Fy	Fz	Fx	Fy	Fz	Fx	Fy	Fz
	(N)	(N)	(N)	(N)	(N)	(N)	(N)	(N)	(N)
Mean	−27	124	318	−200	512	585	−14	35	117
Minimum	−592	−94	−133	−1573	48	−334	−207	−119	−35
Maximum	516	616	1047	1047	1787	1462	176	290	448
Stand. dev.	38.9	33.6	49.0	297.1	270.3	157.5	9.5	21.5	17.3
Stand. error	0.09	0.08	0.11	0.76	0.69	0.40	0.02	0.06	0.04

**Table 6 materials-13-02926-t006:** The results of cutting forces for ap = 2.0 mm, f = 0.1 mm/rev; recorded during turning by tool with a new insert.

Cutting Forces	Fx	Fy	Fz
	(N)	(N)	(N)
Mean	−32	92	374
Minimum	−128	−40	242
Maximum	100	258	515

**Table 7 materials-13-02926-t007:** Average values of chosen parameters recorded during research.

Cutting Parameters		Amplitude	Std. dev.	RMS	Std. dev.	Number of Hits	Std. dev.
ap (mm)	f (mm/rev)	(dB)		(µV)		(-)	
1	0.05	76.0	9.1	32.5	7.5	13,022	3024
1	0.075	63.4	2.6	23.2	3.3	18,724	4477
1	0.1	61.4	6.6	21.1	1.2	9482	3054
1.5	0.05	68.3	0.9	23.0	0.5	8998	4815
1.5	0.075	66.0	5.2	19.4	3.9	8825	3485
1.5	0.1	72.1	9.3	18.6	1.6	10,646	1923
2	0.05	73.2	10.9	25.2	9.2	13,667	4687
2	0.075	85.4	13.8	43.2	21.9	21,649	3954
2	0.1	81.9	4.3	52.8	27.6	14,579	9380
2 (new insert)	0.1	67.7	8.4	18.0	5.8	12,638	12,807

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
