# Peer review of "The Possibility of Applying Acoustic Emission and Dynamometric Methods for Monitoring the Turning Process"

_materials, 2020, doi:10.3390/ma13132926_

Round 1
Reviewer 1 Report
This is an interesting work, using acoustic emission (AE) to optimize the cutting parameters and wear monitoring of the tool. Overall, the manuscript is well-written, but there are some issues. Starting with the minor ones:
- Symbology used to describe the values variability;
- The authors refer several cutting parameters that are not defined. For instance, in line 119, these should be defined at their first appearance. Please revise this issue in order to make it accessible for any reader.
Major issues:
-Figure 16 is not suitable, please create a new plot in a more suitable format.
-Was the data filtered? It is not clear if the authors filtered the forces data;
- The AE method for tool wear monitoring is not new, consider for instance the work of Xiaoli Li, 2002. A brief review: acoustic emission method for tool wear monitoring during turning. International Journal of Machine Tools and Manufacture 42(2), 157-165. https://doi.org/10.1016/S0890-6955(01)00108-0 Please comment on this and make the necessary changes to focus only on the cutting parameters optimization, if that is the case.
- Finally, please be careful with the level of similarity with your previous work https://doi.org/10.5604/01.3001.0012.2785
Author Response
Dear Reviewer,
Thank You for the review. It was very useful for improving our paper.
We added description of symbols used in our work. They were later in the article but we agree that they should be when first time appear in the text.
Fig. 16 was changed.
The data of forces wasn’t filtered – we used raw signals to be sure that every important details were detected. Because the studies were comparative, any errors, distortions connecting with background noise were always the same.
Thank you for very interesting source, Li (2002). We used it in our article. The indicated work is descriptive and general, presenting the state of knowledge for 2002.
Of course we are not first researchers using AE method for monitoring turning processes. We tried to show that it is possible to monitor the process by direct method (dynamometer) and indirect (AE) for obtaining relatively stable conditions of machining related to wear of the tool. We focused on showing exactly parameters for chosen material, tool and cutting parameters. Using AE for the cutting parameters optimization is related with our next work. The sentence about it in the article was our mistake and shouldn’t be placed here. So far, we have completed basic research, which confirmed the sufficient accuracy and sensitivity of the selected method in order to optimize the cutting parameters.
Similarity to our previous work is connected with equipment available in our laboratories – we used the description of them. Thank you for good advice for the future.
Reviewer 2 Report
The study is devoted to the urgent problem of timely diagnosis of cutting tool wear. The authors propose an original technique for using for this purpose the acoustic signal that occurs during the cutting process.
My comments.
1. I am not sure that this article is suitable for this journal by topic. Among the topics covered by this journal https://www.mdpi.com/journal/materials/about there is no study of the mechanical processing of materials.
2. The accuracy with which the authors approach the visualization of their equipment is surprising.
Figure 5 - why? This is a simple microscope. Everyone saw a microscope.
Figure 6 (and a and b) are completely redundant.
And I'm not sure that the previous photos are needed.
3. What important information does table 2 carry? Why compare with certificate data? The conclusions will change greatly if the metal is a little different? Why then there is no data on the composition of the material of the cutter?
The article contains interesting information, and with noticeable processing it will certainly be of interest to specialists. However, I think these are not specialists who constantly read the Materials.
Author Response
Dear Reviewer.
Thank You for your review.
You are right that everybody saw the microscope. We only wanted to show our equipment used in research. In figure 6b is shown sample after spark test. Because chosen method is not NDT, reader can see the damage caused by testing.
Comparing of the chemical composition obtained in research to the manufacturer’s certificate was aimed at confirming that the workpiece is exactly as chosen. In industry there are some cases of differences in the chemical composition of materials supplied under different supplies. A relatively small change in the content of some alloying elements can significantly change the durability of cutting tools during machining. Lack of information about material of the cutter is connected with its construction - the insert is coated so we could only check the composition of outer layer. The tool manufacturer also does not specify the chemical composition of it. At this stage of research, we have not considered this aspect. Thank you for pointing this idea and we will analyze the legitimacy of this topic in further work. The tests should be carried out after using the insert in the cutting zone taking into account individual layers.
Reviewer 3 Report
- Last sentence in abstract’ allowed the determination the beginning of the tool damage process’ should be ‘determination of the beginning’
- In the introduction, the authors state that AE has been used by many researchers to monitor the turning process, and their work is novel because AE is used for cutting parameter optimization. However, I don’t find any optimization throughout the paper. The paper discusses how AE can detect tool wear during turning, but not how to select optimal cutting parameter before turning process.
- Broken language: line 141‘for monitoring the turning process besides cutting forces measuring, was used acoustic emission method.’ Line 164,’ Austenitic steels are the most important group of stainless steels, because have the preferred combination of mechanical properties, machinability and corrosion resistance.
- Why checking the chemical composition of the shaft material? Does it affect the results of AE? There should be a discussion.
- How are the experiments conducted? Are tests in Tables 3, 4, 5 under same insert? It yes, what is the order of the experiment? Does each experiment have the same cutting time or length?
- Resolution of figure 11-13 is low comparing to Figure 14.
- Figure 15, title of plots are unclear
- Figure 16, it is difficult to see the change in frequency from this view. What does different color stand for?
Author Response
Dear Reviewer,
Thank You for the review. It was very useful for improving our paper.
- Language was corrected.
- The works of other researchers, cited in the article, are often descriptive and general, while we focused on the practical aspect - in conditions similar to our research (austenitic stainless steel with a hardness of about 215 HB, a tool with similar geometry characteristic for finish treatment, lack of cooling, etc.).
Using AE for the cutting parameters optimization is related with our next work. The sentence about it in the article was our mistake and shouldn’t be placed here. So far, we have completed basic research, which confirmed the sufficient accuracy and sensitivity of the selected method in order to optimize the cutting parameters. We tried to show that it is possible to monitor the process by direct method (dynamometer) and indirect (AE) for obtaining relatively stable conditions of machining related to wear of the tool. We focused on showing exactly parameters for chosen material, tool and cutting parameters. The novelty is to provide specific RMS values of the AE signal indicating the beginning of rapid wear of the tool, in specific machining conditions, described in the research methodology.
- Broken language was corrected.
- Comparing of the chemical composition obtained in research to the manufacturer’s certificate was aimed at confirming that the workpiece is exactly as chosen. In industry there are some cases of differences in the chemical composition of materials supplied under different supplies. A relatively small change in the content of some alloying elements (i.e.: Titanium, Sulfur, etc.) can significantly change the durability of cutting tools during machining. Changing the turning conditions has a significant impact on the cutting forces and thus on the generated AE signals.
- The tests were carried out using one insert in the following order: ap = 1mm and feed f = 0.05 mm/rev (1-st pin), f = 0.075 mm/rev (2-nd pin), f = 0.1 mm/rev (3-rd pin). Then the test was repeated for ap = 1.5 mm and ap = 2 mm. At the end of the test a turning process was carried out using a new cutting insert - for comparison.
There were the same length of shaft pins – 35 mm.
- Resolution of figures 11-13 was improved.
- Titles of plots were improved.
- Figure 16 was changed. Different colours help to observe changes in amplitude - every 20 dB.
Reviewer 4 Report
The reviewer comments of the paper «The possibility of applying acoustic emission and dynamometric methods for monitoring turning process»
- Reviewer
The authors presented an article «The possibility of applying acoustic emission and dynamometric methods for monitoring turning process». However, there are several points in the article that require further explanation.
Comment 1:
The whole introduction is well written.
However, it will be useful to introduce several articles in the introduction:
https://doi.org/10.1016/j.ymssp.2016.11.026
https://doi.org/10.1007/s00170-019-03607-3
At the end of the introduction, give a brief summary of what has been done in each section before the goal.
Comment 2:
What hardness of the 304L stainless steel workpiece was used?
Comment 3:
Give the geometry of the turning tool. Corners, radii, etc. What is the material of the cutting part? Table with chemical composition, if possible.
Comment 4:
Explain in more detail using cutting physics, a non-uniform change in cutting forces from cutting conditions. Add sentences for this after each figure 7, 8, and 9.
Explain in more detail what this statement was made on line 188: "Significant increasing of radial force indicates a change in geometry of the cutting insert, thus its damage or wear." From figure 9 this does not follow and is not obvious!
Comment 5:
Provide more detailed comments to the figure 10. Check the numbering of the figures in the text.
Comment 6:
Explain in detail in the text the peaks of cutting forces in figures 11, 12 and 13.
Comment 7:
Lines 113, 114 stated:
"This article presents the possibility of using acoustic emission method for optimizing the selection of cutting parameters and wear monitoring of the tool."
However, in Section 3. Results and Discussion and Conclusions, the optimization issue is not explicitly resolved. This decision and explanation need to supplement the relevant sections. The reader and researcher of this method should have a clear idea of how to use it in practice. In addition, the novelty of the article is stated in this. Therefore, this place should be clearly explained.
Comment 8:
The term "turning" should be used in the text of the article instead of "lathing".
Comment 9:
It will be useful to add a section of Nomenclature in which to sign all the physical quantities and abbreviations encountered in the article. There are many physical quantities in the text and such a section will help to find the description of the necessary element.
For example,
ap : Cutting depth (mm)
etc.
Comment 10:
In general, the conclusions are well written, but it is necessary to more clearly show the novelty of the article and the advantages of the proposed method. What is the difference from previous work in this area? Show practical relevance. It is necessary to give quantitative and qualitative indicators of the proposed method. What is the difference from other researchers. The conclusions should be consistent with the purpose of the article.
Comment 11:
The title of the article does not sound convincing enough. It should also clearly express the novelty and originality of the proposed approach in the article.
The article is interesting and relevant, but improvements are needed. After major changes, the article may be considered for publication in journal "Materials".
Author Response
Dear Reviewer,
Thank You for the review. It was very useful for improving our paper.
- Thank you for interesting articles. We added the first one to our article. The second position was used by us before.
- The hardness of the steel was 215 HB.
- The geometry of the tool was added.
C – rhombic insert shape with point angle 80°;
C – insert clearance angle 7°;
E – tolerance (nose height ±0.025 mm; thickness ± 0.025 mm; inscribed circle ±0.025 mm;
T – insert type;
09 - insert size = cutting edge length – 9.525 mm;
T3 - insert thickness – 3.97 mm;
02 –nose radius 0.2 mm;
R –right hand;
MF – chipbreaker for finish turning of stainless steel.
The manufacturer does not specify the chemical composition of the insert. At this stage of research, we have not considered this aspect. Thank you for pointing this idea and we will analyse the legitimacy of this topic in further work. The tests should be carried out after using the insert in the cutting zone taking into account individual layers.
- The process of turning shaft pins with a cutting depth of 1.0 mm and an increased feed value resulted in a steady increase in the force Fz. Increasing the ap parameter to 1.5 mm contributed to a further increase in the analyzed force at 0.05 and 0.075 mm/rev feed. Increasing the feed rate to 1.0 mm / rev allowed to obtain the average value of Fz at the same level, but can be observed is more spread of the results shown in Table 4. This may indicate less stable operation of the cutting tool. Another increase in cutting depth to 2.0 mm at a feed value of 0.05 mm / rev increases the average value of the cutting force, with a large spread of results, as evidenced by the value of the standard deviation (Table 5). Unfavourable cutting conditions which occurred in the previous stage of the study, contributed to the initial degradation of the cutting insert geometry, and during the turning process of the shaft pin with ap = 2.0 mm and f = 0.075 mm/rev of its damage. During the measurement of the force Fz for the cutting process, the difference between the minimum and maximum value of almost 1800 N was registered. Despite the damage to the cutting tool tip, the turning process carried out with an increased feed rate of f = 0.1 mm/rev ensured favourable conditions in the cutting zone and a decrease in the analysed force Fz to the average value. The tests carried out with selected cutting parameters in a similar way affected the values of radial and feed forces. For the cutting depths of 1.0 and 1.5 mm, the turning process was stable, but the increased value of the standard deviation and thus the deterioration of the cutting process conditions are also noticeable. Damage to the tip of the cutting insert caused several increases in the analysed forces Fx and Fy. In the case of Fz, a two-fold increase in value was observed.
- In the first stage of cutting shaft pins (ap = 1.0 mm, f = 0.05, 0.075 and 0.1 mm/rev), only wiping on the surface layer of the cutting insert coating can be observed. The effects of the turning process can be seen only on the rake face, which corresponds to the depth of cut. Increasing the cutting depth to 1.5 mm at the same feed rates results in an additional loss of material at the corner radius and a slight deformation of the major cutting edge in the area of the rake face and flank face. The highest wear of the cutting tool occurred for the process carried out with ap = 2.0 mm. For a feed value of 0.05 mm / rev, a line is visible along the major cutting edge, which may indicate a crack in the insert coating on the chipbreaker. Visible damage in the form of abrasion on the flank face and notch, may contribute to faster abrasion of that surface, resulting in poor surface quality or dimensional inaccuracy of the detail. The insert's wear process started when the feed was increased to 0.075 mm/rev. In the zone of the insert taking active part in the cutting process, the cutting edge was damaged along with breaking the entire corner radius with simultaneous wiping of the rake face and the auxiliary and main flank faces. Continued machining with a damaged insert caused the expansion of the degradation zone as well as thermal overheating and thermal cracks. In addition, micro chipping was observed, which also has a negative effect on surface finish and intensive wear of the flank face.
- In order to properly carry out the turning process of stainless steel, it is important to ensure such cutting conditions, that the chips produced during machining are short or in a controlled manner are discharged into the chip pan of lathe machine. The research was carried out without the use of a cooling liquid, which also contributed to the formation of long chips. In Figures 11, 12, 13, single or periodically repeating peaks can be observed. The formation of a long chip that is not properly removed remains in the direct cutting zone. Jamming of chips between the tool and the workpiece results in the generation of single or periodic increases in forces during the cutting process. This situation may cause the heated chip are cutting again, and the situation may lead to thermal overheating and even chipping of the cutting edge of the tool.
- The sentence about optimization in the article was our mistake. So far, we have carried out basic research, which confirmed the sufficient accuracy and sensitivity of the selected method in order to optimize the cutting parameters. Using AE for the cutting parameters optimization is related with our next work. The novelty is to provide specific RMS values of the AE signal that indicates the beginning of rapid tool wear, in the context of the machining conditions described in the test methodology.
- Lathing was changed into turning in the text.
- Section Nomenclature was added.
Nomenclature
Vc cutting speed [m/min],
f feed [mm/rev],
ap depth of cut [mm],
Fx radial force [N],
Fy feed force [N],
Fz cutting force [N],
Hits number of acoustic emission waveforms exceeding the threshold [-],
A amplitude [dB],
RMS Root Mean Square of the signal [V].
- The works of other researchers, cited in the article, are often descriptive and general, while we focused on the practical aspect - in conditions similar to our research (austenitic stainless steel with a hardness of about 215 HB, a tool with similar geometry characteristic for finish treatment, lack of cooling, etc.).
So far, we have completed basic research, which confirmed the sufficient accuracy and sensitivity of the selected methods. We tried to show that it is possible to monitor the process by direct method (dynamometer) and indirect (AE) for obtaining relatively stable conditions of machining related to wear of the tool. We focused on showing exactly parameters for chosen material, tool and cutting parameters. The novelty is to provide specific RMS values of the AE signal as well as values of forces registered by dynamometer, indicating the beginning of rapid wear of the tool, in specific machining conditions, described in the research methodology.
The use of dynamometers for the analysis of forces during the cutting process makes it possible to observe changeable and unfavourable cutting conditions. The results presented in the article confirm the desirability of using a dynamometer for diagnosing the cutting process. Manufacturers of cutting tools propose a range of cutting parameters optimal for inserts, which should ensure stable cutting conditions with adequate durability. Damage to the cutting edge during the turning process can be observed by analysing the forces during machining the workpiece. Short-term changes in the limit values of registered forces may indicate chip jamming, therefore they cannot be the basis for assessing the condition of the tool. A better solution is observing the average force values. For stainless steel turning with CCET09T302R-MF insert, the indicator of the cutting tool corner damage may be an increase in the average value of the Fz force to the level of 600 N, while increasing the average value of the Fy force to 500 N and Fx to 200 N.
Clear change in RMS of the AE signals (an increase of approx. 100%) could indicate damage to the edge of the turning tool. The maximum RMS value reached 111 µV while turning was performed with a new insert, this parameter was at the level of 20 µV. When turning process was carried out with worn out insert, despite the stabilization of the machining process, the average RMS value did not fall below 40 µV. In case of turning with the worn out insert, the recorded signal was not only characterized by a higher RMS value, but also by greater instability. During turning by worn out tool, the average value of standard deviation of RMS was many times higher, compared to the machining conditions with an unworn tool. This was probably due to the increase in the active length of the cutting edge, and thus the increase in cutting forces. Results allow to conclude that exceeding the RMS value above 40 µV may indicate excessive wear of the cutting tool during machining.
- According to Your advice we propose the title:
Diagnostic parameters of acoustic emission and dynamometric methods for tool wear monitoring during turning process of stainless steel
We haven’t changed it in the article, because we are not sure whether we should complete the entire submission process from the beginning while the title is different.
Round 2
Reviewer 1 Report
Thank you for addressing all my comments. Also, for clarifying that optimization was not carried in this study and it is only planned for future works, establishing thus, a basis for it with this manuscript. However, please revise the introduction, because it is possible to find some sentences where it is indicated that optimization was carried out in this study.
Author Response
Dear Reviewer,
We’ve added two sentences in the abstract and removed one (lines 11-13).
The condition of the cutting tool is one of the factors to achieve this goal. In order to control its wear during the turning process, monitoring was used. In order to optimize the turning process, monitoring is used.
In the introduction information about optimization was removed too (line 86).
and optimization
Thank You one more time for your advices helping us to improving our article.
Reviewer 2 Report
The authors have done significant work to improve the text of the article.
I still have doubts that the authors chose the right journal to publish their results. Since the results obtained are not to materials, their preparation, and properties, but to the methods of machining. But if the authors believe that in this journal the article will find its reader, a decision about acceptability the article for the journal, it remains with the editor.
Author Response
Dear Reviewer,
We chose the special issue of The Materials dedicated to Machining and Manufacturing of Alloys and Steels. In our opinion presented paper should fit here. But, as you said, this is a decision of the editor.
Thank You one more time for your advices helping us to improving our article.
Reviewer 3 Report
Comments are addressed properly
Author Response
Dear Reviewer,
Thank You one more time for your advices helping us to improving our article.
Reviewer 4 Report
All comments are resolved. The article may be published.
Author Response

(The authors gave the same response as above.)
